# Levels and Determinants of Fine Particulate Matter and Carbon Monoxide in Kitchens Using Biomass and Non-Biomass Fuel for Cooking

**DOI:** 10.3390/ijerph17041287

**Published:** 2020-02-17

**Authors:** Zafar Fatmi, Georgia Ntani, David Coggon

**Affiliations:** 1Department of Community Health Sciences, Aga Khan University, PO Box 3500, Karachi 74800, Pakistan; zafar.fatmi@aku.edu; 2MRC Lifecourse Epidemiology Unit, University of Southampton, Southampton SO16 6YD, UK; gn@mrc.soton.ac.uk

**Keywords:** biomass, natural gas, particulate matter, carbon monoxide, kitchen, monitoring

## Abstract

To assist interpretation of a study in rural Pakistan on the use of biomass for cooking and the risk of coronary heart disease, we continuously monitored airborne concentrations of fine particulate matter (PM_2.5_) and carbon monoxide (CO) for up to 48 h in the kitchens of households randomly selected from the parent study. Satisfactory data on PM_2.5_ and CO respectively were obtained for 16 and 17 households using biomass, and 19 and 17 using natural gas. Linear regression analysis indicated that in comparison with kitchens using natural gas, daily average PM_2.5_ concentrations were substantially higher in kitchens that used biomass in either a chimney stove (mean difference 611, 95% CI: 359, 863 µg/m^3^) or traditional three-stone stove (mean difference 389, 95% CI: 231, 548 µg/m^3^). Daily average concentrations of CO were significantly increased when biomass was used in a traditional stove (mean difference from natural gas 3.7, 95% CI: 0.8, 6.7 ppm), but not when it was used in a chimney stove (mean difference −0.8, 95% CI: −4.8, 3.2 ppm). Any impact of smoking by household members was smaller than that of using biomass, and not clearly discernible. In the population studied, cooking with biomass as compared with natural gas should serve as a good proxy for higher personal exposure to PM_2.5_.

## 1. Introduction

Household air pollution from the use of solid fuel (biomass and coal) for cooking or heating has been estimated to cause more than 3.5 million premature deaths per year globally [1,2]. As well as causing respiratory disease and lung cancer [3,4,5], it has been linked with an increased risk of coronary heart disease, although evidence on the latter is less conclusive [6].

Particularly high exposures may occur among women in developing countries who cook on open stoves burning biomass fuels. However, measured concentrations of the two most frequently studied pollutants (particulate matter with aerodynamic diameter <2.5 microns (PM_2.5_) and carbon monoxide (CO)) in cooking areas have varied widely [7,8]. Levels of PM_2.5_ have generally been from 200 to 1000 µg/m^3^, although at the extremes, values <10 µg/m^3^ and >5000 µg/m^3^ have been reported [8]. Measured concentrations of CO have ranged from <1 to >30 ppm [8]. Possible reasons for the heterogeneity include differences in stove design, room configuration and ventilation, and the presence of other sources of pollution such as environmental tobacco smoke.

In Pakistan, where biomass fuels (mainly wood and cow dung) are used for cooking by some 75% of households in rural areas [9], we have conducted a series of studies to explore possible effects on coronary heart disease in women [10,11]. To inform interpretation of those studies, we wished to compare the concentrations of indicator pollutants in kitchens where biomass was used with those in kitchens that used natural gas (a cleaner fuel), and to investigate other factors that might impact on levels of pollution such as the design of the kitchen and stove, and whether there were smokers in the household. To this end, we monitored pollution levels in a random subset of the households that took part in one of the studies of cardiovascular morbidity [11].

## 2. Materials and Methods

The parent study, which had a cross-sectional design, was conducted in villages surrounding the main urban area of Nawabshah district (recently renamed as Shaheed Benazirabad) in the province of Sindh. As described in more detail elsewhere [11], households were recruited through door-to-door visits, to obtain quotas that currently used biomass and natural gas for cooking (Figure 1). We measured PM_2.5_ and/or CO in random subsets of 20 households that used biomass and 19 that used natural gas.

### 2.1. Methods of Measurement

With agreement from the head of the household, fixed site monitors for PM_2.5_ and CO were co-located at a height of approximately one meter (corresponding to the breathing zone of a seated person), approximately one meter from the stove or main fire that was used for cooking. The monitors were attached to a wall or suspended from the ceiling, or where that was not possible, placed on a chair or stool. Care was taken not to site them directly downwind or upwind of the stove/fire.

PM_2.5_ was measured using a single MicroPEM version 3.2 instrument. The device, which was developed by RTI (Research Triangle Institute) International, is lightweight (240 g), portable, and powered by AC mains electricity, but can also operate for 24 h on three AA batteries. Air is sampled through a size-selective inlet that excludes larger particles, and PM_2.5_ is measured with a light-scattering laser photometer, which gives real-time aerosol mass readings. There is also an integrated facility for parallel collection of particulate matter on a filter for gravimetric analysis. In tests conducted outdoors in Durham, North Carolina, over a period during which levels of PM_2.5_ were in the order of 4–33 µg/m^3^, the instrument performed well in comparisons with a Grimm Model EDM180 PM_2.5_ monitor, taken as a reference [12].

For our study, we carried out 48-h continuous monitoring with logging of data at ten second intervals. Where necessary, batteries were replaced every 24 h. All measurements were made before the monitor was due for the first annual recalibration that was recommended by the manufacturer. We also collected ten 48-h parallel samples for gravimetric analysis, but these proved unsatisfactory because filters were inadvertently contaminated. Thus, it was not possible to calibrate the instrument specifically for local particulate matter.

Static monitoring of CO was carried out with a single Q-RAE (version II) monitor over the same 48-h periods, data being logged at thirty-second intervals. The Q-RAE is a small battery-operated device, which weighs about 500 gm and uses an active pump. CO is monitored continuously with an electrochemical sensor, which also responds to several other gases, including hydrogen, ethylene, isobutylene, trichloroethylene, and to a lesser extent, ammonia, hydrogen sulphide, propane, and hexane (cross-sensitivities have been reported for 20 common toxic gases [13]). To increase specificity, the instrument has an in-built oxidizing chemical filter that is designed to remove hydrogen sulphide, and a charcoal (carbon) filter that eliminates most other cross-sensitivities. The oxidizing filter is effective for two years, while under normal operating conditions, the carbon filter needs replacement every 4–6 weeks. However, in our investigation, we used a new carbon filter for each household (i.e., replacement after ≤48 h of measurement), since there was a possibility of exposure to relatively high concentrations of pollutants.

Field workers were trained in the use of the sampling equipment, and a detailed manual with pictorial aids was developed to assist them. In each household, monitoring for PM_2.5_ and CO started at around 10 am. Data were downloaded directly from the MicroPEM and Q-RAE monitors into csv format Excel spreadsheets and text files, respectively.

### 2.2. Determinants of Exposure

Information was noted by the fieldworkers regarding potential determinants of the two pollutants (PM_2.5_ and CO), including the type of fuel (biomass or natural gas), stove (three-stone open traditional stove, improved stove, or gas stove), and kitchen (open or semi-open/closed). In addition, information about smoking in the household was obtained through a questionnaire completed by the woman who cooked in the house (with help from the field worker).

### 2.3. Statistical Analysis

All measurements were collated in two datasets, one for PM_2.5_ and one for CO. After exclusion of those with missing or clearly erroneous values, and correction to zero of those with small negative values (thought to result from minor errors in calibration), arithmetic mean values for each pollutant were derived for each unique combination of household, day, hour, and minute (for PM_2.5_, a mean of up to six logged measurements in the minute, and for CO, a mean of up to two logged measurements). From these minute by minute values, hourly arithmetic mean concentrations were then calculated, together with a count of the number of minutes on which each hourly mean was based. In order to ensure that the hourly means were robust, only those based on 45 or more minutes were retained.

Next, for each household, average concentrations were calculated for each of the 24 h of the day (taking an average from the two days for which measurements had been made if the data were available). Average daily concentrations were then calculated as the means of these hourly average concentrations for households with data on a sufficient number of hours. For two households in which data on PM_2.5_ were missing for only one and two hours respectively, the missing values were imputed using the average value for the relevant time of day in other households with the same type of fuel, and the ratio of average measured concentrations across the other hours of the day to that for all other households using the same fuel. Similar imputation was applied for three households with missing data on CO for one or two hours.

Descriptive statistics were derived for the distributions of hourly mean concentrations in households using biomass and natural gas, and for each type of fuel the average hourly concentration across households was plotted against the time of day.

The relationship between the daily average concentrations of PM_2.5_ and CO across households was examined in a scatter plot, and summarized by a Spearman rank correlation coefficient (rho).

Finally, determinants of daily average fixed site concentrations of PM_2.5_ and CO were explored by multivariate linear regression.

### 2.4. Ethical Considerations

Written informed consent for the installation of fixed site monitoring devices was obtained from the head of each household. The results were shared with the participants, and if they wished, they were advised about possible modifications that might reduce exposures to pollutants at no or minimal cost. The Ethics Review Committee of Aga Khan University approved the study (reference 3119-CHS-ERC-14).

## 3. Results

### 3.1. Completeness of Data

Measurements of both PM_2.5_ and CO were obtained from 18 kitchens using biomass and 19 using natural gas. In addition, CO measurements were made in two further kitchens that used biomass, where for operational reasons it was not possible to monitor PM_2.5_.

The total number of measurements for PM_2.5_ across all households was 744,192 and that for CO 196,618. However, 78,055 measurements of PM_2.5_ were discarded because of clear measurement error (values either missing or <−10µg/m^3^). Among the remaining records (*n* = 666,137), those with marginally negative values from −10 to −1 µg/m^3^, which could plausibly have arisen from minor calibration errors, were set to zero (*n* = 127,275). All measurements of CO were plausible, and none was discarded or amended.

Means were derived for 1826 distinct combinations of household, day, hour, and minute for PM_2.5_, and 1686 for CO. From these, hourly mean values by household were obtained for 859 h for PM_2.5_ and 866 h for CO. However, 19 hourly means for PM_2.5_ and 55 for CO were discarded because measurements were available for <45 min in the hour.

Daily average PM_2.5_ concentrations could not be calculated for two households (one with hourly average concentrations for only 10 h and the other for 12 h). Daily averages for CO were missing for five households, which had hourly averages for 17 h or fewer.

Final analysis was therefore based on 35 households for PM_2.5_ (16 using biomass and 19 natural gas) and 34 households for CO (17 using biomass and 17 natural gas; Table 1). Thirty one households contributed data on both pollutants.

### 3.2. Distribution of Measurements

Hourly mean concentrations of PM_2.5_ and CO in kitchens using biomass were substantially higher than in those using natural gas (Table 2).

In kitchens where biomass was used, the mean of the hourly mean PM_2.5_ concentrations was 531 µg/m^3^ with a median of 136 µg/m^3^. The corresponding values for kitchens where natural gas was used were much lower at 69.9 and 24.2 µg/m^3^_._ For CO, the mean of the hourly mean concentrations in kitchens using biomass was almost twice that in those using natural gas (6.1 vs. 3.4 ppm). However, differences between the median values were smaller (0.8 vs. 0.6 ppm).

Mean hourly average concentrations by time of the day, calculated separately for households using biomass and natural gas, are presented in Figure 2a for PM_2.5_ and Figure 2b for CO. In households using biomass, the highest concentrations of both pollutants were in the evening, with smaller peaks in the morning. In households using natural gas, a similar pattern was apparent for CO, although levels tended to be lower than in kitchens where biomass was used. In contrast, mean hourly average concentrations of PM_2.5_ showed no clear peaks, and were below 180 µg/m^3^ throughout the day.

Daily average concentrations of PM_2.5_ ranged from 59 to 875 µg/m^3^ in households using biomass and from 25 to 172 µg/m^3^ in households cooking with natural gas. For CO, the corresponding ranges were 1.1–17.3 ppm for biomass-using households and 2.1–5.8 ppm for those using natural gas. Figure 3 plots daily average concentrations for CO against those for PM_2.5_ across the 31 households with data on both pollutants. No correlation was observed, either in households using biomass (rho = −0.17, *p* = 0.6) or natural gas (rho = 0.03, *p* = 0.9).

### 3.3. Determinants of Daily Average Concentrations of PM_2.5_ and CO

Table 3 presents results from multivariate linear regression analyses relating daily average concentrations of PM_2.5_ and CO to type of fuel and stove (natural gas stove vs. biomass used with a chimney stove vs. biomass used with a traditional stove), ventilation of the kitchen (closed/semi-open vs. open) and whether there was one or more smoker in the household (environmental tobacco smoke). Effect estimates for the above factors and their 95% confidence intervals (CIs) were mutually adjusted in each of two models, one for PM_2.5_ and one for CO.

In comparison with households using natural gas for cooking, PM_2.5_ concentrations were significantly higher in those that used biomass, with either a chimney stove (mean difference 611, 95% CI: 359, 863 µg/m^3^) or traditional three-stone stove (mean difference 389, 95% CI: 231, 548 µg/m^3^). Open kitchens tended to have lower PM_2.5_ concentrations than closed/semi-open kitchens (mean difference −88.3, 95% CI: −325, 148 µg/m^3^), and smoking in the house was associated with higher PM_2.5_ levels, although not significantly (mean difference 84.5, 95% CI: −65.4, 235 µg/m^3^).

For CO, concentrations were significantly increased with use of biomass in a traditional stove (mean difference from natural gas 3.7, 95% CI: 0.8, 6.7 ppm), but not with the use of biomass in a chimney stove (mean difference −0.8, 95% CI: −4.8, 3.2 ppm). Smoking in the house was associated with higher CO, although not to the point of statistical significance. There was no major difference in concentration of CO according to whether kitchens were open or closed.

## 4. Discussion

Our results indicate that in kitchens using biomass for cooking, average airborne concentrations of CO, and especially PM_2.5_ were higher than in those using natural gas. Use of a chimney stove appeared to reduce levels of CO, but not of PM_2.5_. Any effects of smoking on the levels of pollutants were smaller and not clearly discernible.

The average levels of PM_2.5_ in houses using biomass were some 50–70-fold higher than standards for ambient air in western countries. They are consistent with other studies that have measured PM_2.5_ in kitchens of biomass-users [8], and indicate a potential for high exposure among women who cook in such kitchens, where they would be expected to spend at least 2–3 h per day, often at times when levels exceeded 1000 µg/m^3^. Concentrations were highest for the longest duration during the evening when most household members, including the women, would be at home.

Levels of CO were also higher in kitchens using biomass, but unlike PM_2.5_ were lower when a chimney stove was used. The difference may have contributed to the lack of correlation between daily average concentrations of PM_2.5_ and CO across kitchens. There were also indications that cooking with gas produced CO but not PM_2.5_. Thus, the levels of both pollutants varied substantially by the time of the day in houses using biomass fuel, with two peaks, which most likely corresponded with the main times of cooking. For CO, peaks in concentrations seemed to occur at the same times of the day in the kitchens using natural gas. Previous studies support the possibility that appreciable quantities of CO are generated when natural gas is used for cooking [14,15].

Use of biomass fuel appeared to be the main determinant of pollutant concentrations in kitchens. Even where biomass was used with a chimney stove, levels of PM_2.5_ were clearly elevated and similar to those associated with traditional three-stone stoves, whereas CO concentrations were close to levels in houses using natural gas. This suggests that stoves with a chimney tend to reduce CO but have little influence on PM_2.5_. Most studies have found that when well-designed standardized chimney stoves were introduced (in intervention trials), they reduced levels of PM and CO [7,16,17], although in one investigation, neither PM_2.5_ nor CO was significantly lower in kitchens with homemade chimneys [18]. These apparent inconsistencies may reflect differences in the design of chimneys and levels of ventilation in the kitchens studied, and further research is needed to confirm which designs of chimney are most effective.

Smoking in the household tended to be associated with higher concentrations of PM_2.5_ and CO, but any effects on levels of the pollutants appeared to be much smaller than those of using biomass for cooking. It was expected that the levels of pollution produced by the burning of biomass fuel would be high compared with those from ETS since few women in the study area were active smokers and any contribution would be mainly from smoking by men, who culturally did not spend much time in kitchens.

No clear reduction in pollutant levels was apparent in open as compared with more closed kitchens. The kitchens in the households studied varied in size, type, construction material and ventilation levels, and the distinction between open and closed kitchens may not always have been clear-cut.

### Potential Limitations

Mainly because of technical problems with equipment, the number of households studied was fewer than planned, which limited the power to compare different types of kitchen, as described above. However, even with the reduced sample size, large differences between fuel types were clearly apparent.

Although households were randomly selected for air monitoring from those participating in the survey of cardiovascular morbidity, the latter were recruited by quota sampling. However, there is no obvious reason why the study sample should have been systematically unrepresentative with regard to differences in pollution levels according to whether solid fuel was used for cooking.

The MicroPEM measures fine particles at concentrations ranging from 1 to 10,000 mg/m^3^ [19], and performed well when tested against a gravimetric standard [12]. Due to a contamination problem, we were unable to validate our continuous monitoring against gravimetric analysis of samples collected in parallel. Therefore, the absolute values of the PM_2.5_ measurements may not be fully accurate, and should be interpreted with caution. However, it seems unlikely that measurement error could account for the large diurnal variation in PM_2.5_ concentrations that we observed only in kitchens using biomass, or the much larger differences in pollution levels that were associated with use of biomass than with type of kitchen or smoking in the household.

The main limitation of the QRAE monitor was the possibility of cross-sensitivity to other gaseous pollutants. The instrument was designed to minimize this problem, but the possibility remains that measurements were somewhat inflated by other pollutants, and this may have contributed to the absence of correlation between daily average concentrations of CO and PM_2.5_.

Some measurements were missing because of equipment failure and temporary problems with electrical supply. However, analyses were based only on households with sufficient measurements to characterize exposure levels reliably.

When the study team visited households to install monitoring equipment in the kitchens, they directly observed the type of fuel, type of stove, and type of kitchen. Therefore, the classification of those variables should have been accurate. Smoking in the household was reported by the woman who cooked in the house, and should also have been ascertained fairly accurately.

## 5. Conclusions

Our study found substantially higher average concentrations of CO and particularly PM_2.5_ in the kitchens of biomass-users, which is consistent with other studies. Ventilated kitchens tended to have somewhat lower levels of the pollutants, and houses with smokers somewhat higher concentrations. However, stove chimneys as used in the kitchens studied, had no discernible impact on levels of PM_2.5_. It follows that within that population, the type of fuel used for cooking can be considered a good index of potential for substantially higher personal exposures to the pollutant among women who cook, with the type of stove, type of kitchen, and smoking in the household being less influential.

## Figures and Tables

**Figure 1 ijerph-17-01287-f001:**
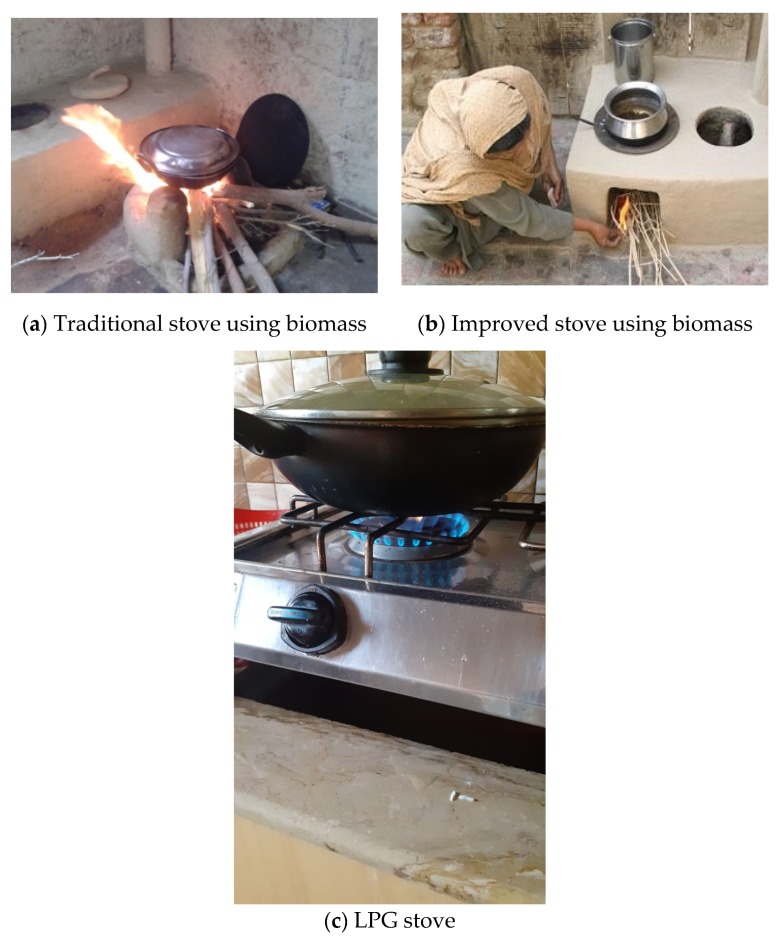
Methods of cooking in the study population.

**Figure 2 ijerph-17-01287-f002:**
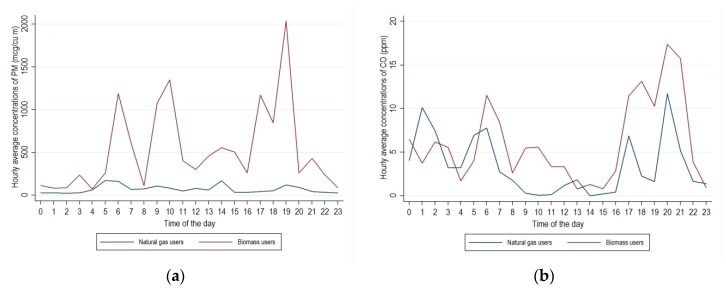
Arithmetic means across households of hourly average concentrations of (**a**) PM_2.5_ (µg/m^3^) and (**b**) CO (ppm) by time of day and type of fuel.

**Figure 3 ijerph-17-01287-f003:**
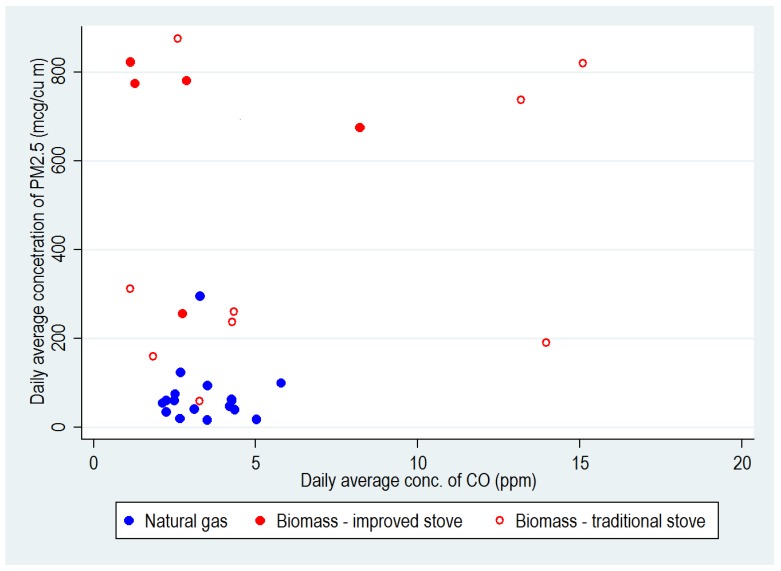
Scatterplot of daily average concentrations of PM_2.5_ and CO in households using biomass and natural gas for cooking (based on 31 households with satisfactory data on both pollutants).

**Table 1 ijerph-17-01287-t001:** Completeness of data.

	**PM_2.5_**	**CO**
**Households monitored**		
Using biomass	18	20
Using natural Gas	19	19
**Number of measurements across all households**		
Total	744,192	196,618
Discarded because of clear measurement error	78,055	0
Retained	* 666,137	196,618
**Combinations of household, day, hour, and minute for which mean concentrations were derived**	1826	1686
**Hours for which hourly mean values by household were derived**		
Total	859	866
Discarded because measurements available for <45 min in the hour	19	55
Retained	840	811
**Daily Average Concentrations could not be derived because data missing for >2 h**	2	5
**Daily average concentrations derived and used in analysis**		
Households using biomass	16	17
Households using natural gas	19	17

***** Includes 127,275 marginally negative values corrected to zero.

**Table 2 ijerph-17-01287-t002:** Distribution of hourly mean concentrations of PM_2.5_ and CO in kitchens of households using biomass and natural gas for cooking.

	PM_2.5_ (µg/m^3^)		CO (ppm)	
	Biomass	Natural Gas	Biomass	Natural Gas
Mean	531	69.9	6.1	3.4
Minimum	4.2	4.2	0	0
Maximum	4930	2580	92.0	35.5
Median	136	24.2	0.8	0.6
25th percentile	34	13.5	0	0
75th percentile	615	53.3	6.4	4.9
90th percentile	1650	147	16.0	11.2

**Table 3 ijerph-17-01287-t003:** Mutually adjusted multivariate linear regression coefficients for factors that might influence PM_2.5_ and CO concentrations in kitchens. Analysis is based on 35 households for PM_2.5_ and 34 for CO.

Risk Factor	Mean Difference in Daily Average PM_2.5_ Concentration with 95% CI (µg/m^3^)		Mean Difference in Daily Average CO Concentration with 95% CI (ppm)	
Natural gas stove	Reference		Reference	
Biomass with chimney stove	611	(359, 863)	−0.8	(−4.8, 3.2)
Biomass with traditional stove	389	(231, 548)	3.7	(0.8, 6.7)
Closed/semi-open kitchen	Reference		Reference	
Open kitchen	−88.3	(−325, 148)	0.6	(−3.1, 4.4)
Environmental tobacco smoke (ETS)				
No	Reference		Reference	
Yes	84.5	(−65.4, 235)	2.1	(−0.7, 5.0)

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
