# Peer review of "Levels and Determinants of Fine Particulate Matter and Carbon Monoxide in Kitchens Using Biomass and Non-Biomass Fuel for Cooking"

_ijerph, 2020, doi:10.3390/ijerph17041287_

Round 1

Reviewer 1 Report

The paper addresses an important topic that can affect the health of large number of people, especially women in rural areas. However, there are some serious shortcomings and comments that need to be addressed. These comments are indicated in the attached pdf file.  

Reviewer 2 Report

I was interested in effect of the types of stoves. Figure 2 could identify not only fuels but also types of stoves. In the Discussion, the subheading of 4-1 should be 4-2, and add 4-1 such as “main findings and suggestions”

Reviewer 3 Report

This manuscript presents the results of measurements performed in the kitchens of families using biomass and natural gas. The presentation is clear and well documented, and the results are consistent with the cited literature. I would like to ask the authors to clarify the following aspects:

1) should we refer to PM2.5 measurements to qualitative data due to the lack of gravimetric analysis?

2) Why did the authors not perform measurements outside, potentially relevant for closed kitchens, to verify the possible contribution of ambient air concentrations? Were the experimental sites similar regarding the environmental levels of PM2.5 and CO?

3) If I understand correctly from Table 2, PM2.5 levels are higher in kitchens that use fireplaces compared to traditional ones. This result is difficult to understand since the fireplaces are used to reduce pollution in indoor environments. Can the authors clarify this point?

Finally, I recommend accepting the work after a possible clarification of these aspects.

Round 2

Reviewer 1 Report

Please see the red text in the attached file.
